# Comparing Donor- and Acceptor-Originated Exciton Dynamics in Non-Fullerene Acceptor Blend Polymeric Systems

**DOI:** 10.3390/polym13111770

**Published:** 2021-05-28

**Authors:** Chan Im, Sang-Woong Kang, Jeong-Yoon Choi, Jongdeok An

**Affiliations:** Department of Chemistry, Konkuk University, 120 Neungdong-ro, Gwangjin-gu, Seoul 05029, Republic of Korea; tony22@konkuk.ac.kr (S.-W.K.); haru96@konkuk.ac.kr (J.-Y.C.); anhijuk@konkuk.ac.kr (J.A.)

**Keywords:** polymeric photovoltaics, non-fullerene acceptors, transient absorption spectroscopy, exciton dynamics, internal quantum efficiency

## Abstract

Non-fullerene type acceptors (NFA) have gained attention owing to their spectral extension that enables efficient solar energy capturing. For instance, the solely NFA-mediated absorbing region contributes to the photovoltaic power conversion efficiency (PCE) as high as ~30%, in the case of the solar cells comprised of fluorinated materials, PBDB-T-2F and ITIC-4F. This implies that NFAs must be able to serve as electron donors, even though they are conventionally assigned as electron acceptors. Therefore, the pathways of NFA-originated excitons need to be explored by the spectrally resolved photovoltaic characters. Additionally, excitation wavelength dependent transient absorption spectroscopy (TAS) was performed to trace the nature of the NFA-originated excitons and polymeric donor-originated excitons separately. Unique origin-dependent decay behaviors of the blend system were found by successive comparing of those solutions and pristine films which showed a dramatic change upon film formation. With the obtained experimental results, including TAS, a possible model describing origin-dependent decay pathways was suggested in the framework of reaction kinetics. Finally, numerical simulations based on the suggested model were performed to verify the feasibility, achieving reasonable correlation with experimental observables. The results should provide deeper insights in to renewable energy strategies by using novel material classes that are compatible with flexible electronics.

## 1. Introduction

Recently, non-fullerene type acceptors (NFA) have gathered remarkable attention owing to the rapid increase in their photovoltaic power conversion efficiency (PCE) [1,2,3]. One of the most important reasons for the impressive PCE improvement must be the extended absorption spectral range, which can provide higher short circuit current densities (J_SC_) as expressed by Equation (1) with additional solar cell parameters, e.g., open circuit voltage (V_OC_) and fill factor (FF). And P_in_ and P_out_ in Equation (1) stand for the incident solar power and the converted output power, respectively. Obviously, the absorption facility of an active layer must be a determining factor at the initial primary exciton (PE) forming stage of solar cell operation. However, for instance, charge transfer (CT) and charge carrier (CC) transport must be considered together because the photogenerated PEs must be converted to mobile CCs and extracted to external circuitry to be detected as J_SC_ [4]. Furthermore, most events occur in the solid states, such as CT and CC transport, and, therefore, J_SC_ must also strongly depend on their molecular morphology [5]. Nevertheless, this spectral extension is achieved by using noble NFA materials coupled with suitable polymers as a bulk heterojunction (BHJ) active layer. The benefits of NFA systems are clearly recognized by comparing with solar cells that comprise fullerene derivatives and, for instance, narrow bandgap polymers [6] or polythiophene derivatives [7].

(1)
$$\mathrm{PCE}=\frac{P_{\mathrm{out}}}{P_{\mathrm{in}}}=\frac{J_{\mathrm{SC}}\times V_{\mathrm{OC}}\times \mathrm{FF}}{P_{\mathrm{in}}}$$


One way of further increasing the PCE of NFA-based solar cells involves chemical tailoring via the halogenation of NFA materials; for example, fluorinating ITIC (~11% PCE) [8,9] to ITIC-4F (~13% of PCE) [10]. The fluorination needs to be applied not only to the NFA material but also to the corresponding donor polymer, often referred to as PBDB-T [6], to form the fluorinated derivative, PBDB-T-2F [10]. The chemical names and structures of the abovementioned fluorinated NFA and polymeric materials are presented in Appendix A. NFA fluorination has triggered intense re-search activity to achieve higher PCE, demonstrated by the graphical trend of the “Best Research-Cell Efficiency Chart” [11] and insightful reviews [12,13]. Further advancements besides chemical tailoring include the use of ternary BHJ systems [14,15,16], in which multiple NFA components are blended together to form an active layer for single junction photovoltaic cells, and the use of tandem multijunction device structures [17]. Simultaneously, efforts are continued to further enhance the spectral absorption range of active materials [18,19].

The aforementioned research has managed to depict the underlying mechanism of photovoltaic power conversion of fluorinated derivatives, e.g., the CC photogeneration, CC transport through the bulk layer, and the CC extraction across various interfacial layers. The initial CC photogeneration is of particular significance because it also occurs in the extended spectral range of the noble NFA-based BHJ layer to achieve a higher PCE. Because the initial CC photogeneration process is often bound to ultrafast CT from PEs to preferable CC generation species, the transient absorption spectroscopy (TAS) was used as an inadvertent tool to explore those behaviors [20,21,22,23]. Based on the results of these studies, the perspectives of NFA-based photovoltaic devices and related materials were extensively discussed [24,25,26].

However, the behavior of the PEs originating from NFA materials in a blend solid system remains largely unclear although the light harvesting spectral ranges of NFAs were remarkably extended, and, therefore, significant amount of incident photons must be captured by the NFAs. The contribution of the PEs originating from an NFA to the PCE, via excitation at wavelengths below the polymeric donor’s absorption edge, can reach almost ~30% in the case of the PBDB-T-2F:ITIC-4F blend system. Surprisingly, the most part of the pure NFA absorption band arose due to their film formation but not in their isolated mutual states as in diluted solutions. Therefore, the PBDB-T-2F:ITIC-4F blend system provide a chance to investigate the pathways of the NFA origin and donor origin excitons in blend systems separately by using a site-selective TAS method despite complex superimposing of various relevant excited species simultaneously. The complicated superimposing in addition to various interactions, e.g., CTs, between active species of the blend systems can be further clarified by comparing them with the virtually unperturbed exciton dynamics of the diluted solutions and solidification-induced exciton dynamics of the pristine film samples in a successive manner. The changes accompanied by film formation might be discussed in terms of more specific aggregation, for instance, three-dimensional molecular stacking with preferable orientation, ordering including crystallinity issues must be one of determinant factors [27,28]. However, the aggregation related factors were not explicitly considered in this work, instead, the changes accompany with film formation was treated as an average ensemble behavior within a framework of more general solidification, i.e., three-dimensionally dense packed situation where there is pronounced diffusivity of excitons play a significant role. Combining information about the exciton behaviors of blend solid systems, pristine solid systems, and diluted solutions in conjunction with their photovoltaic properties within blend systems can provide crucial scientific insights of the blend solid states which can contribute for further improvement of those types of photovoltaic devices.

In this study, the PBDB-T-2F:ITIC-4F blend and their pristine film samples are investigated by the spectrally resolved absorptance, incident photon to current conversion efficiency (IPCE), and internal quantum efficiency (IQE) to estimated their steady-state PE forming characters with their corresponding device properties. The experimental findings indicate clearly that the NFA, ITIC-4F, can play a role as donors, in particular at the longer wavelength range, although they are conventionally assigned as acceptors. In addition, a series of site-selective TAS was performed to understand the nature of the excited states originating from an NFA in addition to those originating from a donor polymer separately. Site-selective excitation with proper wavelengths, 565 nm and 695 nm, was used not only for the blend film but also for the solutions and pristine films to identify the origins and pathways of the excited states in the complex blend system. Unique site-selective decay behaviors were observed in addition to the drastic changes of the decay behaviors of the solutions to pristine samples upon solidification. By using the obtained results, an origin-specific reaction kinetics model that was assisted by the photoluminescence quantum yield (PLQY) values was suggested. Finally, the proposed model was verified through numerical simulations showing analogous results to experimental observables.

## 2. Materials and Methods

Measurements were performed using fluorinated PBDB-T-2F and ITIC-4F purchased by 1-Material Inc. Hereafter, the fluorinated PBDB-T-2F and ITIC-4F are denoted as **P** and **I**, respectively. Their chemical structures and names are shown in Appendix A. UV-Visible (UV-VIS) absorption and photoluminescence (PL) spectra of their solution samples were obtained from solution concentrations of about 1 × 10^−4^ mg/mL in chlorobenzene (CB) solvent. Pristine films of those fluorinated materials, in conjunction with their blending film, were spun cast on quartz substrates at a 50:50 wt% blending ratio for spectroscopic characterization. The film preparation conditions were kept consistent with those of active layer films in devices to enable direct comparison between spectroscopic and device properties.

The blended active layers and ZnO thin layers were spun cast onto indium-tin-oxide (ITO)-covered glass substrates; then, molybdenum oxide/silver (MoO_3_/Ag) counter-electrodes were thermally evaporated. [8,10] The obtained device structure was as follows: glass (1.1 mm)/ITO (180 nm)/ZnO (40 nm)/**P**:**I** BHJ active layer (100 nm)/MoO_3_ (18 nm)/Ag (80 nm). The thickness of the active layers was measured using a surface profiler (XP-200, AMBiOS). All devices were characterized according to their parameters, i.e., open circuit voltage (V_OC_), short circuit current density (J_SC_), fill factor (FF), and PCE, via current density-voltage (J-V) measurements under standard illumination conditions. The test condition deployed is the standard 1-sun, often referred to as Air Mass 1.5-G spectral irradiance, at a power of 1 kW/m^2^ [6]. The IPCE and IQE were also estimated using XE-7 (PV Measurement Inc.).

A two-beam UV-VIS spectrometer (Neosys-2000, Scinco) was used to obtain UV-VIS absorption spectra of solution samples. A fiber-coupled spectrometer (AvaSpec-ULS2048, Avantes) was used to obtain reflectance and transmittance spectra of film samples which can be used to form absorptance spectra by using the following straightforward relation that the total sum of absorptance, reflectance, and transmittance at a given wavelength must be unity. Additional reflectance spectra were measured for devices with opaque metal electrodes to calculate IQE spectra. Steady-state PL spectra both for the solution and films were measured using a spectrophotometer (FS-2, Scinco), and absolute PLQYs were obtained using a commercial system (Quantaurus-QY, C11347, Hamamatsu Photonics K.K.) equipped with an integrating sphere.

Ultrafast TAS was performed by utilizing a Ti:sapphire regenerative amplifier laser system (Libra, Coherent) with a pulse width of approximately 100 fs at a repetition rate of 1 kHz in conjunction with a pump–probe spectrometer (Helios, Ultrafast Systems LLC). To select a specific excitation wavelength, an optical parametric amplifier system (TOPAS, Coherent) was used. A sapphire crystal was implemented for white light continuum to probe the excited species. The TAS kinetics were measured up to about 2 nanoseconds (ns) with picosecond (ps) time-resolution, and the probing wavelength ranges covered most of the VIS (550–850 nm) and IR (900–1400 nm) bands. In most TAS cases, excitation beam intensities of about 10 μW were used with the average pump beam size of approximately 1 mm in diameter, intending to reduce intensity-dependent bimolecular behaviors by decreasing the photon flux density. The TAS spectra were recorded as the change in optical density, (∆OD) using a personal computer and software provided by the manufacturer of the pump–probe spectrometer.

## 3. Results

### 3.1. Absorptance, IPCE, IQE, and PCE

The UV-VIS absorptance spectra of solution samples, pristine films, and blend films of **P** and **I** are shown in Figure 1. The arrows mark the excitation wavelengths of 575 nm (solid line) and 695 nm (dashed line) required for TAS to excite the **P**-rich and **I**-rich spectral regions, respectively. Approximately 70% of the incident photons with a 575 nm wavelength were absorbed by **P**, while virtually all incident photons with a 695 nm wavelength were absorbed by **I**. It is notable that the absorption band of **I** exhibits significant redshift compared to **P**, when the materials were formed as pristine films. The redshift of **P** from solution to pristine film was virtually negligible, but that of **I** was approximately 0.36 eV; the redshift was computed from the difference between absorptance edges from the 673 nm of solution to the 810 nm of pristine film. The absorptance edge and spectral shape of **I** were not significantly affected upon blending with **P**. The selected excitation wavelength for the **I** film could be longer to ensure selective excitation of **I**. However, the value of 695 nm was chosen to excite the **I**-rich spectral range of both solution and film samples to enable the consistent comparison of exciton behaviors.

Figure 1B shows the IPCE spectrum obtained by a photovoltaic device using a **P**:**I** blend film as the active layer. The corresponding IQE spectrum is also shown after proper absorptance correction according to Equations (1) and (2). The IPCE values at 575 nm and 695 nm are comparable (73% and 70%, respectively), although the former wavelength selectively excites the **P**-rich spectral range, while the latter selectively excites the **I**-rich spectral range. It is noteworthy that the most incident photons can be captured by **I** at the **I**-rich range because there is no significant absorption of **P**, therefore, the “**I**-rich” range can be considered practically as the “**I**-only” absorbing spectral range. After proper correction via the internal absorptance of the **P**:**I** active layer, the difference in IPCE was less pronounced in the IQE spectrum. Specifically, the IQE spectra of the studied blend film device was approximately 91%, as observed in the flattened plateau region of Figure 1B. This indicates that both types of excitons comparably contributed to the PCE, although the primary excitons generated at different wavelengths differ and are expected to undertake distinct decay paths.

(2)
$$\mathrm{IPCE}=\frac{\#\;\mathrm{of}\;\mathrm{total}\;\mathrm{extracted}\;\mathrm{CCs}}{\#\;\mathrm{of}\;\mathrm{total}\;\mathrm{incident}\;\mathrm{photons}}$$


(3)
$$\mathrm{IQE}=\frac{\#\;\mathrm{of}\;\mathrm{total}\;\mathrm{extracted}\;\mathrm{CCs}}{\#\;\mathrm{of}\;\mathrm{total}\;\mathrm{absorbed}\;\mathrm{photons}\;\mathrm{in}\;\mathrm{an}\;\mathrm{active}\;\mathrm{layer}}$$


A summary of the PCE and related device parameters, i.e., V_OC_, J_SC_, and FF, of the studied solar cells is shown in Table 1. The device used in this study had a 50 wt% blending ratio and a typical PCE of approximately 11.5%. In addition, the **P** pristine film devices exhibited a PCE of 0.19% while the PCE of the **I** ones were only 0.02%. This may imply that the **P**:**I** blend film as an active layer in bulk heterojunction type devices comprises additional pathways contributing to the PCE and achieves a 2~3 orders of magnitude higher PCE.

In Figure 1C, a spectral irradiance spectrum [29] is presented as a reference; the spectral irradiance spectrum was multiplied by the fraction number of the IPCE spectrum to account for the incident solar power captured by the device. The resulting spectrum shows that the contribution to the PCE by the absorptance of the **I**-rich spectral region is approximately 30% of the whole spectral range of the **P**:**I** blend film device. When considering the whole photon capture ratio achieved by **I** regardless of the spectral ranges, the ratio may reach ~50% being comparable to the blend ratio. That seems reasonable at a glance, but one must confront with some discrepancies as soon as one assumes that all PEs are generated only on **P** as an electron donor conventionally. In particular, the photocurrent generation below the wavelength of ~650 nm range can only be expected by the excited electron transfer from PEs of **I** because only **I** can absorb photons at that spectral range. Therefore, it is crucial to explore the pathways of PEs depending on their origin, especially for **I** which is still often assigned as a non-fullerene acceptor in most literatures.

### 3.2. Exciton Dynamics of Solution and Pristine Film Samples

The IR range transient absorption (TA) spectra of solutions **P** and **I** are shown in Figure 2(A1,A2), respectively.

The PE band of **P** (hereafter, PE(**P**)) has a much broader shape than the PE band of **I** (hereafter, PE(**I**)), which can be observed in the ∆OD spectra sliced at various time delays, as shown in Figure 3A,B.

Furthermore, PE(**I**) decays virtually monoexponentially with a spectroscopic lifetime (τ_0_) of ~392 ps while PE(**P**) follows a multiexponential decay law. The fitting operation of the kinetic traces for the selected wavelength was performed by using Equation (4). This allows to take into account the full width at half maximum of instrumental response function (IRF) by a sum of convoluted exponentials which has crucial meaning to analyze the kinetics especially near the prompt region.

(4)
$$\Delta \mathrm{OD}\left(t\right)=e^{-{\left(\frac{\tau -\tau_{0}}{\mathrm{IRF}/\left(2\cdot \ln2\right)}\right)}^{2}}*\sum_{i}A_{i}e^{-\left(\frac{\tau -\tau_{0}}{\tau_{i}}\right)},$$

where τ_0_, A_i_, and τ_i_ stand for time zero, amplitudes, and decay times, respectively. The symbol * in Equation (4) is to express the convolutional operation to the following term. The fitting parameters of P and I solutions are presented in Appendix A as an example. In Table 2, dominant spectroscopic lifetimes of crucial states are summarized. The kinetic TAS trace of the PE(**P**) solution fit the trace at an acceptable level after more than three spectroscopic lifetimes (see inset of Figure 3B)—especially at the tail part of the trace). To fit the kinetic trace at the full measurement time range of ~3 ns, at least 2–3 additional lifetimes were required, implying that the kinetic trace of the PE(**P**) is gradually extended and follows a stretched exponential, i.e., Kohlrausch-Williams-Watts type decay function [30,31,32].

The broad PE band width and the stretched exponential decay behavior of the PE(**P**) solution must be explained by a polydisperse molecular weight, i.e., broad chain length distribution typical of such polymeric materials. [32] Compared to oligomeric **I** with a well-defined molecular weight and a smaller size, the PE(**P**) on polymer chains should be able to relax energetically. Therefore, the decay behavior of the PE(**I**) solution can be described by homogenous unimolecular recombination, while that of the PE(**P**) solution should exhibit serious dispersive effects owing to the pronounced molecular weight distribution and diffusivity on the distributed chains. In addition, possible bimolecular behavior of PE(**P**) solution was traced by performing a series of excitation intensity dependent TAS for sure. In Appendix A, three normalized kinetic traces measured with excitation intensities of 7 μW, 20 μW, and 40 μW are shown together with a calculated mono-exponential decay curve by using a lifetime of 650 ps. With decreasing excitation intensity, i.e., decreasing PE(**P**) population on polymer chains, their spectroscopic lifetimes were extended, and their functional shapes were converging toward the calculated mono-exponential decay pattern. Consequently, the spectroscopic lifetime of 650 nm was taken as an unperturbed intrinsic decay parameter to describe excitons on the isolated polymer chains with a low population density. The lifetime to describe the PE(**P**) solution with other lifetimes of relevant excited species are listed in Table 2. Although the topics related to the detailed exciton behaviors on polymer chains are important and must be further investigated, these issues are not continued to discuss at this stage because the discussion may exceed the current scope of this study.

Drastic changes could be observed upon solidification of the materials even in the absence of blending both **P** and **I**, as shown in the TAS spectra of their pristine films of Figure 2(B1,B2). The lifetimes of PE(**P**) and PE(**I**) of pristine film samples were shortened approximately 2 orders of magnitudes compare to those of solution samples and listed values as τ_p1_ in Table 2. It is clear to recognize that the shortening of lifetimes was triggered by the formation of new solid-state bands. [33,34] The efficient population from PEs to the newly formed states might be explained by means of conventional charge transfer (CT) states. This notion is reasonable because the materials, **P** and **I**, are already proved as excellent active materials for photovoltaic devices where they must have efficient CT nature against competitive dissipation to regenerate their mutual GSs. Therefore, it is noticed that the newly formed bands are referred to as CT bands hereafter, despite their practical complexity to avoid any possible misconducting. Subsequently, the new bands originated from the **P** and **I** of pristine films were denoted as CT(**P**_solid_) and CT(**I**_solid_), respectively. In addition to the lifetime τ_p1_, the subsequent decay lifetimes of the new CT population which had approximately 1–2 orders of magnitudes longer than those of τ_p1_ are listed in Table 2 as τ_p2_.

Although the redshift of **P** was negligible, the reduction in PL quantum yield (PLQY) during solidification was remarkable, as shown in Table 3.

The PLQY of **P** decreased almost an order of magnitude from ~11% to ~0.9%. Contrarily, the PLQY of **I** exhibited a slight decrease from ~6.8% to ~6.1%, despite the pronounced redshift observed in the absorption spectra of the **I** pristine film. Interestingly, the different PLQY reduction trends upon solidification of **P** and **I** seem comparable with the different lifetime shortening trends of them. The shortening of CT(**P**_solid_)’s lifetime was pronounced as the effective PLQY reduction in **P** pristine film, while the shortening of CT(**I**_solid_)’s lifetime was marginal being comparable to the slight PLQY reduction in **I** upon solidification. The **P**:**I** blend film exhibited the lowest PLQY value of ~0.3% as expected due to the high PCE of the photovoltaic cells using **P**:**I** blend active layer. This means that the PLQY of **I** decreased significantly upon blending, while that of **P** was not seriously affected. The blending effects will be presented with TAS kinetics in the following Section 3.3.

Prior to the move to Section 3.3, the relevant energy levels of **P** and **I** collected with TAS of solution and pristine film samples combined with the previously reported energy values of the highest occupied molecular orbitals in square brackets [10] are presented in Figure 4.

The less perturbed energy difference (ΔE) values between PEs and ground states (GS) were extracted from the absorption edges of the UV-VIS spectra of the solution samples, while solidification-induced ΔE values were extracted from those of the solid-state pristine film samples. Actually, the band shapes and positions of pristine films’ UV-VIS spectra were not significantly changed upon blending of the materials. Therefore, the **P**:**I** blend solid-state systems must be handled as a mixture of the intrinsic **P** and **I** states in addition to the newly formed solidification-induced **P** and **I** states. This means that the four significant decaying components shown in Figure 2(B1,B2) are appeared in the kinetic traces of the **P**:**I** blend system concurrently overlapping to each other. This superimposing feature hinders to identifying the dynamics within blend system separately, if the kinetic components are not successively obtained and compared by using the solution, pristine film, and blend film samples, although the energy levels may not be significantly changed. It is noticed that the newly formed bands upon film formation are expressed with a more general term, solidification, rather than a more specific one, aggregation, to avoid expanding the study with, for instance, detailed molecular orientation and crystallinity issues even though they are crucial to understand the photovoltaic solid systems. The ΔE values between PE and higher lying transition bands detected in the TAS of the pristine films were extracted from the maximum peak of the corresponding TA spectra (shown with angle brackets). Because of the different estimation method of ΔEs for higher transitions, the values might be slightly overestimated compared with the values extracted from the band edges.

### 3.3. Exciton Dynamics of P:I Blend Systems

In Figure 5, the TA spectra of a **P**:**I** blend film with excitation wavelengths of 575 nm and 695 nm are shown. These excitation wavelengths were chosen to selectively generate the PEs of either **P** with 575 nm or **I** with 695 nm. The TA spectrum of the PE(**P**) generated with an excitation wavelength of 575 nm (see Figure 5(A1)) exhibits a characteristic broad band of **P** pristine film previously observed in Figure 2(B1) and Figure 3A. However, there was a superimposing feature for the PE(**I**) centered at ~960 nm because of the overlapping of both **P** and **I** absorption bands at 575 nm. Consequently, the PE(**P**) and PE(**I**) had to be generated simultaneously at 575 nm, as shown in Figure 1, although the absorptance ratio between **P**:**I** was about 1:0.44. The amplitude ratio between the GS **P** and GS **I** bands (hereafter, GS(**P**) and GS(**I**), respectively) extracted from the prompt maximum part of the visible range TA spectrum was also 1:0.38. The negative ΔOD of the TA spectra should correspond to the GS photobleaching bands, as shown in Figure 5(A2) (blue color mapping). Finally, in Appendix A), the ratio can be easily estimated from the negative ΔOD spectrum being sliced at a time delay of 0 ps. In Figure 5(B2), the visible range of the TA spectra of the blend film excited at 695 nm is presented to depict the excitation wavelength dependent depopulation of the GS bands and their successive recovering behaviors. For the case of the 695 nm excitation, the ratio between **P**:**I** bands near the prompt maximum TA spectrum is ~1:1.8, while the corresponding ratio of the 575 nm excitation is 1:0.38. The spectra with time delays of 0 ps, 10 ps, and 100 ps are shown in Appendix A for clear comparison. Site-selective kinetic traces of the GS photobleaching bands are also shown in Appendix A. The negative values of spectra were converted to positive to plot them with a double logarithmic scale.

The ratio of 1:0.38 for 575 nm excitation case coincides with the ratio obtained from the steady state absorptance spectra. Contrarily, the ratio of 1:1.8 for 695 nm excitation case cannot be easily explained by the superimposed absorption of **P** and **I** because the PE(**P**) cannot be directly generated at 695 nm regardless of the weak absorbing band tail. Despite the relatively broad spectral width of ~10 nm due to the ultra-short laser pulse length of ~100 fs, a relatively high portion of the GS photobleaching band mutually belonging to **P** cannot be easily explained. This implies that the PE(**I**) excited at 695 nm could be converted to **P**-related CT bands extremely fast. [21,22] Therefore, a strong interaction between **P**_solid_ (possibly **P** also) and PE(**I**) is suggested, particularly when the PE(**I**) dominates over the PE(**P**). These distinctively different excitonic behaviors of CT(**I**_solid_) and CT(**P**_solid_) in the **P**:**I** blend film are analogous to the uniquely different PLQY changes upon blending. The PLQY of **I**, i.e., radiative decay facility, was seriously affected while that of **P** was not significantly reduced. This remarkable difference of the blend system is further discussed using energy level diagrams and reaction path schemes in the Discussion section.

The TAS spectra of the pristine and blend films shown in Appendix A allows the comparison of the different spectral decaying behaviors of the PE(**P**) and PE(**I**). The prompt TAS spectra of the PE(**I**) in the pristine film and **P**:**I** blend film were virtually same at a time delay of 0 ps; however, their spectral shape changed for time delays over ~10 ps. For short delays, a band centered at ~950 nm became dominant in a **P**:**I** blend film, while it almost completely dissipated in the **I** pristine film. This indicates that most of the CT(**I**_solid_)^−^ in the **P**:**I** blend film can be transferred to other species, which are newly integrated into **I** by blending with **P**. This newly integrated species effectively interact with the CT(**I**_solid_)^−^, probably including the residual CT(**I**)^−^ after solidification. This provides a plausible explanation about why the PE(**I**) remains similarly PL-radiative in the pristine film upon solidification but not in the **P**:**I** blend film. Contrarily, the PE(**P**) in the pristine and **P**:**I** blend films has shown a drastically different prompt TAS spectra at a time delay of 0 ps. However, the difference should be caused by the overlapping of both absorption bands at 575 nm, i.e., the simultaneous direct generation of PE(**P**) and PE(**I**), although the interaction in this case did not seem very effective.

Summing up, unlike PE(**I**), the PE(**P**) is not effectively affected by the blending process. It is noteworthy that the conventional fullerene blend systems, e.g., PTB7:PC71BM [32] and P3HT:PC61BM, [7] have shown a systematic reduction in the lifetimes of PE originated from polymeric donors upon blending with fullerene acceptors. This difference can be recognized more clearly by exploring the kinetic decay traces extracted from the three-dimensional TAS spectra of various samples, as shown in Figure 2 and Figure 4. Figure 6A shows the normalized ΔOD traces of the **P** pristine film. The wavelengths of 908 nm and 1147 nm correspond to the spectral regions where the kinetic traces were taken for the PE(**P**) and CT(**P**_solid_), respectively. By comparing these kinetic traces with those of the PE(**P**) in solution, a similarity is observed between the secondary solidification-induced decay and the mutual PE(**P**) dynamics. 

Additionally, the PE band traced at ~908 nm decayed much faster than the secondary band traced at ~1147 nm. The lifetime of the PE band was approximately an order of magnitude shorter than that of the secondary band, baring remarkable similarities to the reduction trend of the PLQY upon solidification. However, the decay of CT(**I**_solid_) at ~1297 nm of **I** pristine film is not significantly shorter than its mutual decay obtained in **I** solution when this relation is compared with that of P case despite the change of the decay function of I from a mono-exponential exponential type to a stretched exponential type as shown in Figure 6B). Again, those distinctive decaying behaviors between **P** and **I** can be comparable with the of PLQY differences between **P** and **I** as aforementioned. Similarity between **P** and **I** systems is clearly recognized by the efficient depopulation from their PE bands due to the formation of new solidification-induced bands. Figure 6C) shows the normalized ΔOD traces of the **P**:**I** blend film. The kinetic traces were extracted at wavelengths near 950 nm and 1300 nm. The decay of the energetically higher lying bands is generally rapid but remarkably slower than that of the pristine film cases. Furthermore, the decay kinetics of the energetically lower lying bands appear significantly dependent on the excitation wavelength. The mutual decay character of PE(**P**) in the pristine film is reasonably preserved in the selective generation of PE(**P**), while ~31% of the PE(**I**) character is overlapped by the nature of the PE(**P**); this could be considered in terms of the relative insensitive relation between PE(**P**) and PE(**I**). However, the kinetic characters of PE(**I**) were preserved only at the beginning of in the selective generation of PE(**I**), within a time range of approximately 10 ps. Then, the decay character was dominated by the integration effects of **P**.

## 4. Discussion

Based on the identified energy levels and estimated spectroscopic lifetimes, kinetic reaction paths were suggested, as shown in Figure 7. Although the spectra and kinetic traces of the **P**:**I** blend film samples were complex owing to the successive and simultaneous decay of various excited species, plausible spectroscopic lifetimes could be extracted in a successive way. First, the mutual unimolecular decay lifetimes of PE, τ_0_, were estimated using the TA spectra of solution samples. Then, the lifetimes, τ_p1_, of the electron transfer from PE to newly formed CT in pristine films were estimated. Moreover, the lifetimes τ_p2_ were characterized by the depopulation rate of the newly formed CT of pristine films. As a final step, the lifetimes τ_b1_ for the charge transfer toward neighboring CT(**P**_solid_) from CT(**I**_solid_)^−^ were estimated; similarly, the depopulation lifetimes τ_b2_ of CT(**P**_solid_)^−^ were estimated, as listed in Table 2. Because the PE(**P**) was not seriously affected by the blending with **I**, **P** in blend system related lifetimes, τ_b1_ of τ_b2_ were not explicitly estimated. Similarly, a reaction scheme related to **P** was suggested only until the pristine film scenario, where the PE(**P**) was severely influenced by the newly formed CT(**P**_solid_) as shown in Figure 7B. Contrarily, a reaction scheme related to **I** was proposed until the blend system scenario as shown in Figure 7A.

The extracted lifetimes and suggested processes were verified by a numerical simulation based on the described reaction kinetic model using a set of differential equations. Equation 3. Equations (4), and (5) are coupled differential equations based on the reaction schema for the PE(**I**) case shown in Figure 7A. Equations (6) and (7) correspond to the PE(**P**) case; this case has less terms than the PE(**I**) case owing to the relatively weak interaction in the blend film. The rate constants, k_0_, k_p1_, k_p2_, k_b1_, and k_b2_ for the simulation are reciprocal values of the estimated spectroscopic lifetimes, τ_0_, τ_p1_, τ_p2_, τ_b1_, and τ_b2_, respectively. For the simulation, the initial population of PE was expressed as a Gaussian function with a full width at half maximum of 1 ps and unit area to account for the laser pulse duration with average instrument response function (IRF). The simulation results indicate that the suggested model is trustworthy for describing the exciton decay in complicated **P**:**I** blend systems, by comparing the simulation results in Figure 8 and experimental results in Figure 6. 

The simulated kinetic traces for the PE(**I**) seem to recover the most critical characteristics of the experimental data on an acceptable level, confirming the validity of the suggested kinetic model when PE(**I**) is dominant. However, the simulated results of the PE(**P**) in the **P**:**I** blend solid system did not seem to achieve impressive results. One of the possible reasons might be the fact that the numerical simulation was not explicitly considered about the spectral relaxation bound with a stretched exponential decay pattern of the PE(**P**). However, the stretched exponential behavior including bimolecular aspects could not consider keeping the scope of the current study for better overview.

(5)
$$\frac{d{\left[\mathrm{PE}\left(I\right)\right]}_{t}}{\mathrm{dt}}=G_{t}-k_{0}{\left[\mathrm{PE}\left(I\right)\right]}_{t}-k_{p1}{\left[\mathrm{PE}\left(I\right)\right]}_{t}$$


(6)
$$\frac{d\left[\mathrm{CT}{\left(I_{\mathrm{solid}}\right)}^{-}\right]}{\mathrm{dt}}=k_{p1}{\left[\mathrm{PE}\left(I\right)\right]}_{t}-k_{p2}{\left[\mathrm{CT}{\left(I_{\mathrm{solid}}\right)}^{-}\right]}_{t}-k_{b1}{\left[\mathrm{CT}{\left(I_{\mathrm{solid}}\right)}^{-}\right]}_{t}$$


(7)
$$\frac{d\left[\mathrm{CT}{\left(P_{\mathrm{solid}}\right)}^{-}\right]}{\mathrm{dt}}=k_{b1}{\left[\mathrm{CT}{\left(P_{\mathrm{solid}}\right)}^{-}\right]}_{t}-k_{b2}{\left[\mathrm{CT}{\left(P_{\mathrm{solid}}\right)}^{-}\right]}_{t}$$


(8)
$$\frac{d{\left[\mathrm{PE}\left(P\right)\right]}_{t}}{\mathrm{dt}}=G_{t}-k_{0}{\left[\mathrm{PE}\left(P\right)\right]}_{t}-k_{p1}{\left[\mathrm{PE}\left(P\right)\right]}_{t}$$


(9)
$$\frac{d\left[\mathrm{CT}{\left(P_{\mathrm{solid}}\right)}^{-}\right]}{\mathrm{dt}}=k_{p1}{\left[\mathrm{PE}\left(P\right)\right]}_{t}-k_{p2}{\left[\mathrm{CT}{\left(P_{\mathrm{solid}}\right)}^{-}\right]}_{t}$$


What excited species are mainly responsible of achieving the relatively high IQE of 91% in the **P**:**I** blend film device on both **I**- and **P**-rich spectral regions remains to be answered. To properly address this question, the difference between the TAS and operating device conditions regarding excitonic dissipation pathways must be considered. For TAS, a single layer thin film on a quartz substrate is typically used to obtain reliable kinetic spectra of the bulk film without severe artifacts. Therefore, all excitonic processes triggered by a shot of ultrafast laser pulses should be halted by a thin interior film that can recover mutual GS. Furthermore, every circulation must be finished between the pulses to avoid serious accumulation effects owing to, for instance, long-living triplet-related species [35] and immobilized space charges [36], which can affect the landscape where the exciton process occurs.

Contrarily, the operating device conditions have particularly different aspects compare to TAS. First, the active layer is intercalated between buffer layers, such as ZnO and MoO_3_ combined with transparent conducting oxide and opaque metal electrodes. Second, photovoltaic devices are typically used and characterized under the continuous illumination of light. Lastly, those devices supply electric power on to a load through externally bound circuitry. Consequently, precursors of CC in an operating device do not recombine to become mutual GSs to liberate the absorbed energy within the bulk film. Precursors of CC can transform to free mobile CC either in the bulk or at the interfacial layers just before leaving the bulk layers to deliver the absorbed energy to the external electric load. Additionally, these processes must happen under quasi steady-state equilibrium, when there are more trapped or mobile CC-like species in the **P**:**I** bulk heterojunction film. This should cause the landscape where excited species interact enormously different from non-equilibrium dynamic single shot conditions for TAS.

In addition to the charged species within the sample, interactions at the interfacial layers might also occur, causing significantly different behaviors than when they are in a bulk layer on a relatively inert glass or quartz substrate. Complexity should increase when considering interference between forward- and reverse-propagated beams passing through the active film owing to opaque reflecting metal electrodes. [6] Continuous operating situations combined with external circuitry can significantly impact the total reaction kinetics. If the CCs are quickly extracted before they are dissipated, via unimolecular or bimolecular processes, which can be a further driving force to produce CCs in the bulk. When mobility through bulk or escaping across the interfacial layer are rate-limiting steps, the CC generation should be reduced; otherwise, the generation process within the sample must be accelerated [37]. Thus, the CC generation must be considered in conjunction with the interfacial layers and external circuitry to estimate the effects on the kinetic processes inside the active layer.

Nevertheless, the proposed model suggests that the CT(**P**_solid_)^−^ and its counterpart, GS(**I**)^+^, might be practical precursors to become free mobile CCs, when the PE(**I**) is dominant as illustrated in Figure 7A. The population of CT(**P**_solid_)^−^ reaches up to approximately 93% according to the numerical simulation under the described single shot kinetic condition. This means that the system suffers an inadvertent loss of approximately 7% owing to, e.g., relatively fast unimolecular recombination. A similar instinctive loss in a narrow-band-gap polymer and fullerene blend system was previously reported [38]. Contrarily, when the PE(**P**) is dominant, the situation is different owing to the insensitive behavior of **P** upon the incorporation of **I** and/or **I**_solid_. Therefore, the most plausible species to become mobile CCs are CT(**P**_solid_)^−^ and GS(**P**)^+^ for PE(**P**) case. However, the yield of mobile CCs from **P** might be comparable to that of **I** because of the efficient CT from the mutual PE(P) to the newly formed CT(**P**_solid_) generated by solidification. Consequently, the reason for the low PCE of **P** pristine film device can be answered with the idea that the photogenerated CCs, i.e., electrons and holes, cannot extracted from the active layer effectively due to the imbalance of the transporting channels suitable for each CC type. [39] Therefore, **P** pristine films must be assisted by coexistence of **I**. which proved by the high PCE of **P**:**I** blend systems.

## 5. Conclusions

At least ~30% of the total PCE could be contributed by the PE(**I)** formed at the **I**-rich spectral range with a device comprising a **P**:**I** blend active layer. The extended spectral range of **I** appeared by solidification, where there is no absorption by **P** suggests that **I** must be able to play a role as a donor. The excitation wavelength selective TAS shows that PEs with different origins were distinctively different both spectrally and kinetically although their facilities to contribute toward PCE are similar. In addition to the dramatical change upon solidification, the PE(**I**) was significantly affected by the incorporation of **P** during the decaying process of the **P**:**I** blend, while the PE(**P**) was not. An origin-specific reaction kinetics model was suggested by using TAS data and verified by a numerical simulation, which was reasonably comparable to experimental kinetic data.

## Figures and Tables

**Figure 1 polymers-13-01770-f001:**
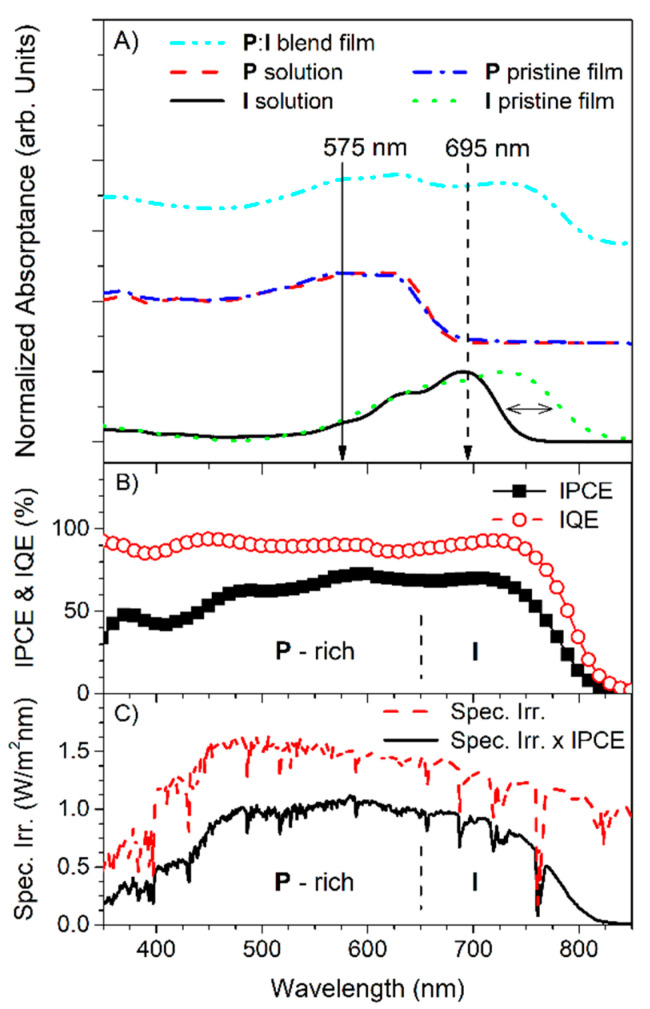
(**A**) Absorptance spectra of a **P**:**I** blend film, **P** pristine film, **I** pristine film, **P** solution, and **I** solution. Arrows mark the excitation wavelengths of 575 nm (solid line) and 695 nm (dashed line). The percentage absorptance is plotted rather than the logarithmic absorbance to enable direct comparison of the IPCE and IQE spectra; (**B**) IPCE spectrum (rectangular markers) and corresponding IQE spectrum (circular markers) of the **P**:**I** blend film; (**C**) Spectral irradiance (Spec. Irr.) spectrum (dashed line) of the global tilt solar radiation under the air mass 1.5 condition [29] and its weighted spectrum (solid line) are multiplied by the IPCE spectrum of (**B**).

**Figure 2 polymers-13-01770-f002:**
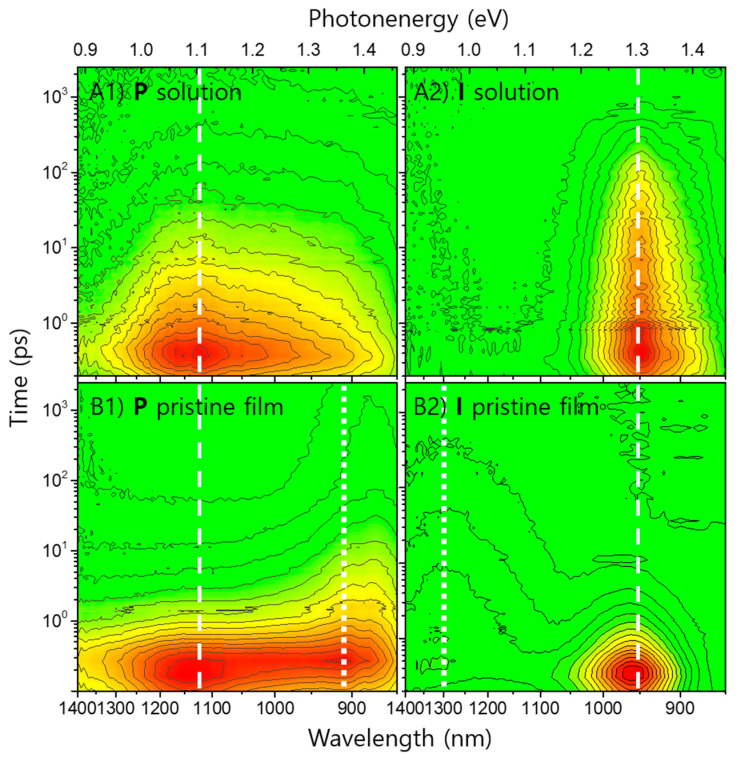
IR range TA spectra of (**A1**) **P** solution, (**A2**) **I** solution, (**B1**) **P** pristine film, and (**B2**) **I** pristine film. The excitation wavelength is 575 nm. The ∆OD intensities are mapped with linear-scale contour lines combined with color assignment from positive values (red) to zero (green). Additional white dashed and dotted lines mark significant TA band positions.

**Figure 3 polymers-13-01770-f003:**
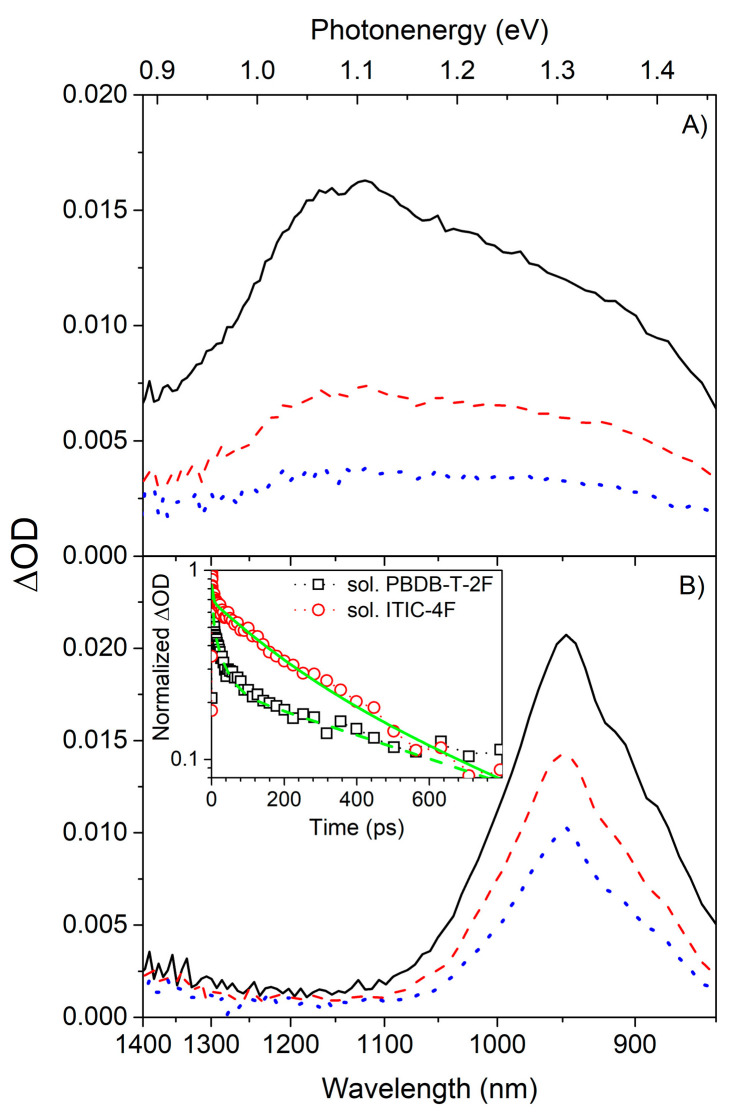
ΔOD spectra measured with solutions of (**A**) **P** and (**B**) **I**, at an excitation wavelength of 575 nm, and extracted at various delay times of about 0 ps (solid line), 10 ps (dashed line), and 100 ps (dotted line) with decreasing ΔOD amplitude. In the inset of (B), two kinetic traces of **P** (open rectangles) and (**B**) **I** (open circles) selected at the maximum wavelengths are shown with multi-exponential fitting curves.

**Figure 4 polymers-13-01770-f004:**
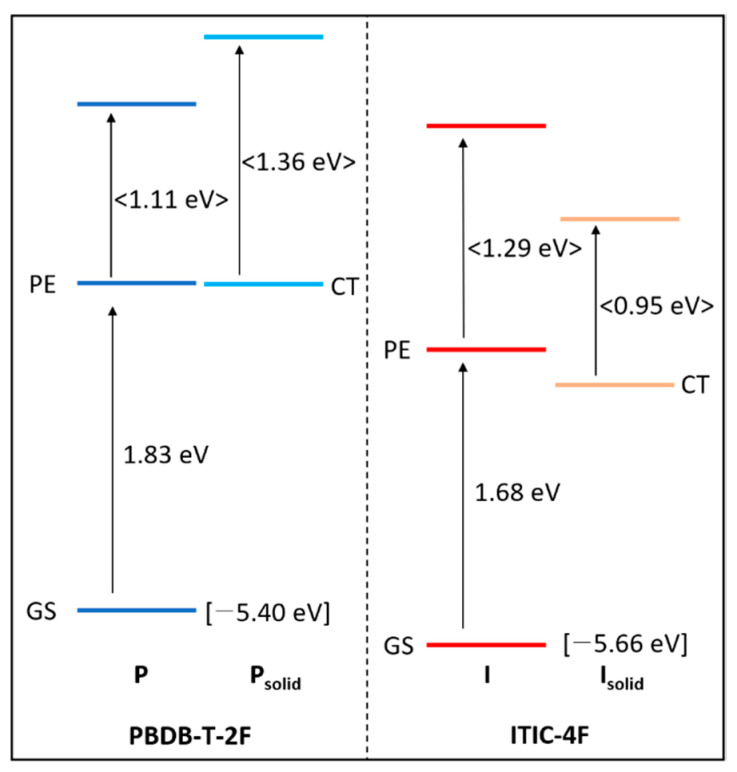
Energy level diagrams of **P** and **I** materials.

**Figure 5 polymers-13-01770-f005:**
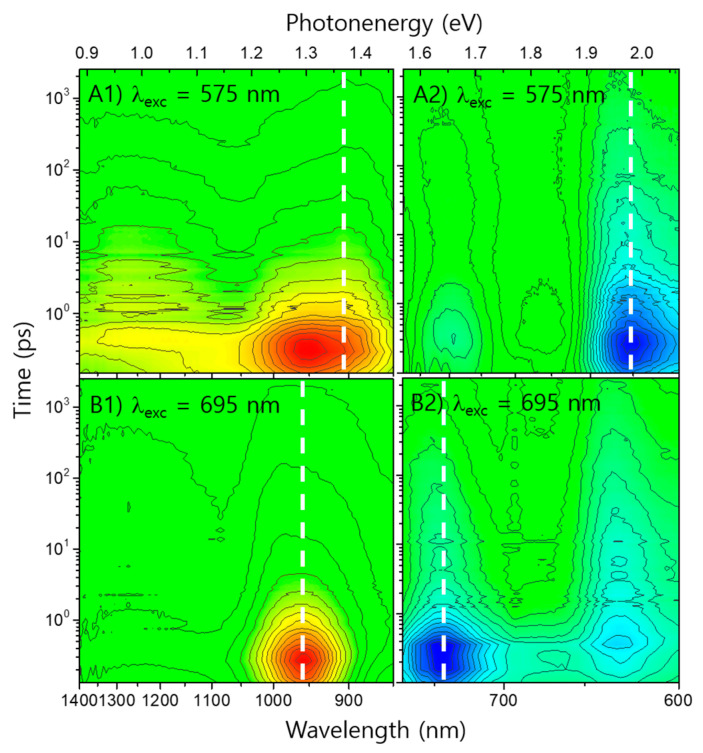
TA spectra of **P**:**I** blend films excited at 575 nm for the (**A****1**) IR and (**A****2**) visible range in addition to those excited at 695 nm for the (**B****1**) IR and (**B****2**) visible range. The ΔOD intensities are mapped with linear-scale contour lines combined with color assignment from positive values (red) to zero (green) to negative values (blue). The white dashed lines mark the main bands of pristine film samples.

**Figure 6 polymers-13-01770-f006:**
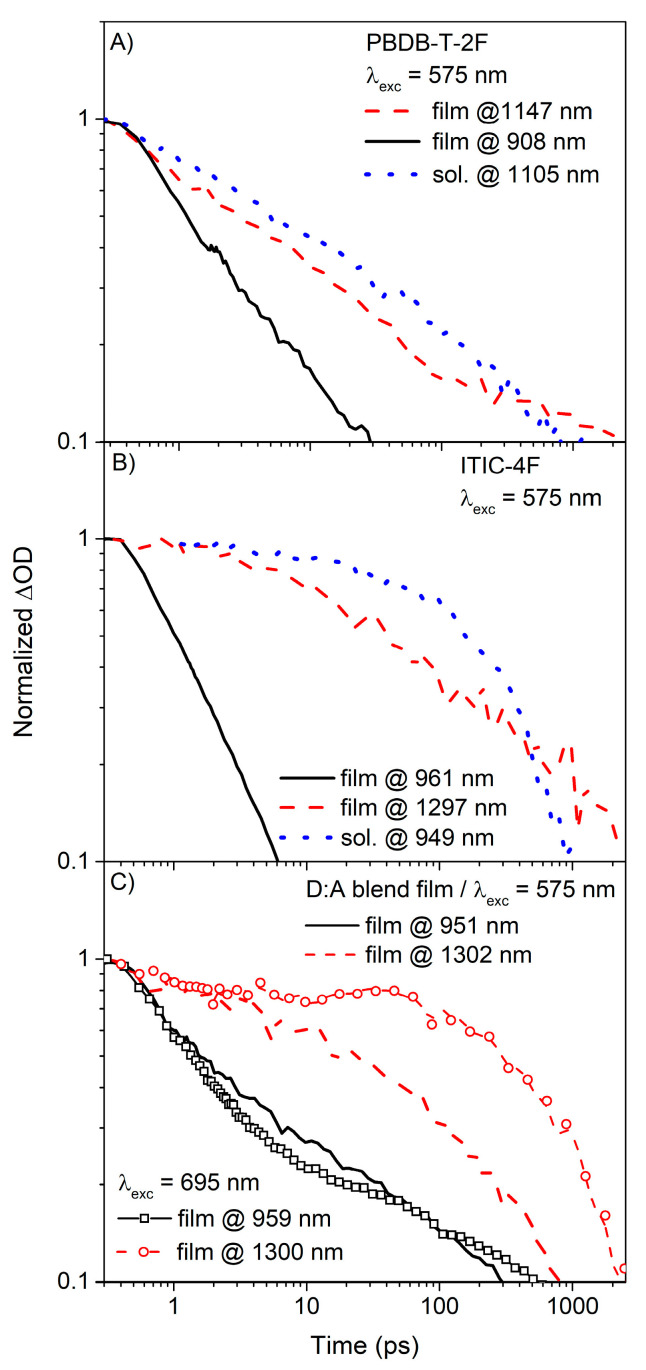
Normalized ΔOD kinetic traces of the (**A**) **P** pristine film, (**B**) **I** pristine film, and (**C**) **P**:**I** blend films. For (**A**,**B**), the excitation wavelength was 575 nm. For (**C**), the kinetic traces of ΔOD with excitation wavelengths of both 575 nm and 695 nm (curves with markers) are shown for comparison.

**Figure 7 polymers-13-01770-f007:**
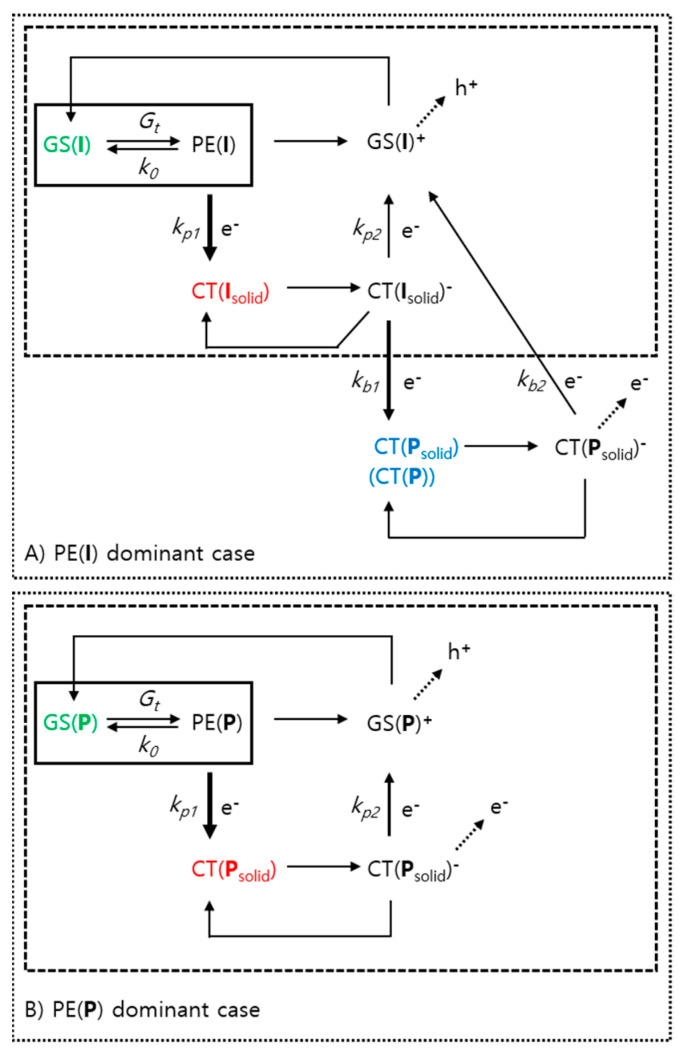
Decay paths of (**A**) PE(**I**) and (**B**) PE(**P**) as reaction schema. The reactions in boxes with solid lines correspond to solution samples, while the boxes with dashed lines correspond to additional reaction paths after film forming. Boxes with dotted lines correspond to additional dissipation paths in cooperation with **P** in blend film samples.

**Figure 8 polymers-13-01770-f008:**
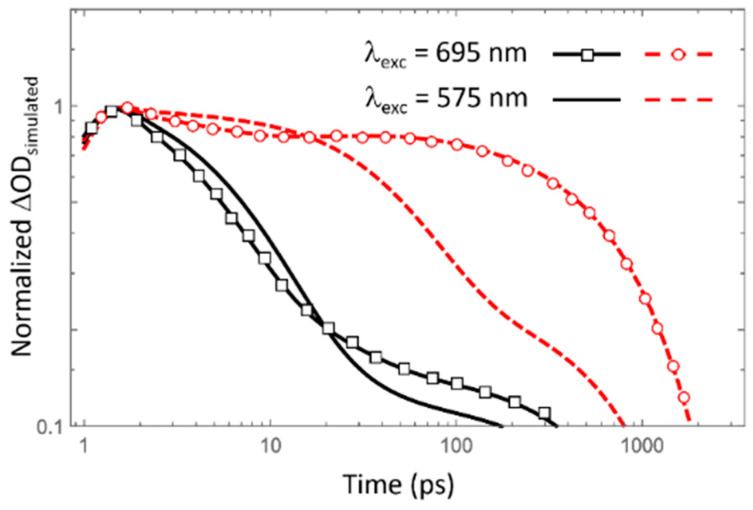
Numerically simulated ΔOD for different excitation wavelength cases. The black solid and the red dashed lines show the simulated decay kinetic traces for energetically higher lying bands at ~950 nm and lower lying bands at ~1300 nm, respectively.

**Table 1 polymers-13-01770-t001:** Overview of device parameters.

	PCE [%]	V_OC_ [V]	J_SC_ [mA/cm^2^]	FF [%]
**P**:**I** blend	11.5	0.85	20.1	67
**P** pristine	0.19	0.69	0.32	83
**I** pristine	0.02	0.17	0.26	36

**Table 2 polymers-13-01770-t002:** Dominant spectroscopic lifetimes in picosecond time unit.

	τ_0_	τ_p1_ and τ_p2_	τ_b1_ and τ_b2_
[ps]	solutions	pristine films	blend films
PE(**I**)	392	4/450	30/850
PE(**P**)	650	7/45	-/-

**Table 3 polymers-13-01770-t003:** PLQY of studied samples.

PLQY [%]	I	P	P:I
Solution	6.8	11.0	-
Film	6.1	0.9	0.3

## Data Availability

The data presented in this study are available on request from the corresponding author.

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
