# Peer review of "Comparing Donor- and Acceptor-Originated Exciton Dynamics in Non-Fullerene Acceptor Blend Polymeric Systems"

_polymers, 2021, doi:10.3390/polym13111770_

Round 1
Reviewer 1 Report
This paper presents a ‘Comparing Donor- and Acceptor-Originated Exciton Dynamics in Non-Fullerene Acceptor Blend Polymeric Systems’ by Chan Im et al for understand the mechanism of Non-fullerene type acceptors solar cells. The title is falling into the field of chemistry, physics and energy materials, and the studied contents include the preparation, characterization and testing of materials, such as absorptance spectra, PL, IPCE, IQE and device parameters as well as dynamics informations. The research results provide a certain understanding of the micro mechanism of NFA system. But the manuscript needs to be revised to improve the overall quality of the article by revision, and specific suggestions are as follows:
- In the Abstract of this manuscript, too many experimental methods are introduced, but the main conclusions and innovations are not introduced. It is suggested to add this part.
- introduction, the sentence ‘The impressive PCE improvement is mainly attributed to the extended absorption spectral range, which can provide higher short circuit current densities (JSC) in addition to the increase of open circuit voltages (VOC).’ is not accurate. indeed, absorption spectral range could affect JSC; however, many factors are influencing Jsc such as charge transfer and transport as well as morphology, as the latest reference ‘low-bandgap porphyrins for highly efficient organic solar cells: materials, morphology, and applications’ and ‘modification of NFA-conjugated bridges with symmetric structures for high-efficiency non-fullerene PSCs’. So corresponding factors affecting the photoelectric performance of Jsc and Voc should be introduced.
- why did the author choose these two molecules (PBDB-T-2F and ITIC-4F) as the research objects?
- In 3.1, how did the UV-Vis absorption spectra of the sample films detect by using the conventional means? What is the difference between this method and the result of the solution sample test?
- For Fig.1a, the absorption spectra of P solution and film have no significant difference, but that of I solution and its corresponding film has the apparent change; what are the reasons? The authors also explain the cause of the redshift of the absorption spectrum of thin-film, such as for ITIC-4F two phases.
- In 3.2, for the obtained dynamic parameters, what the fitting model used in the experiment? The author needs additional explanation.
- From the time life obtained, the author should get more detailed information of these two samples.
- Compared Fig.B1 with Fig.A1, it can be found there is an additional peak in the P pristine film; please explain the reason for this phenomenon.
- For the Fig.4, the author indicated the energy levels of P and I were according to Ref.8, is there any better value for this work?
- There are some unclear English expressions and grammatical errors in this paper:For example, a) in line 177-180 ‘The IPCE values at 575 nm and 695 nm are …while the latter selectively excites the I-rich, better to say, almost I-only, spectral range’, the expression of this sentence is not clear. It is suggested to sketch it again; b) ‘Complexity increases as soon as one must consider …. phenomenon for such diluted solution’ the sentence should be rewritten; c) ‘There are more aspects which must be further investigated, however further discussion related to the detailed exciton behaviors on polymer chains is restricted to focus on the scope of this study’ the sentence should be rewritten; d) ‘This means that the blend system has all the components shown in Figure 4 instantaneously which makes the dynamics within blend system seriously complicated although the energy levels may not be changed.’ the sentence should be rewritten.
Author Response
We sincerely appreciate the valuable comments given by the reviewer with the expense of the reviewer's time and efforts.
Please see the attachment.
Thanks a lot in advance.
Chan Im

Reviewer 2 Report
This is an interesting study of Im and coworkers, focusing on an interesting area called photovoltaics. While the work is nicely done, I have a few suggestions for its improvement.
1- The conclusion section is too long. It should be reduced to a maximum of 5-6 lines.
2-Tle last paragraph of Introduction must be rewritten, as the objectives discussed of this paragraph is unclear.
3-As it seems, the authors conducted only experiments and reported their results. However, their data are not supported by theoretical calculations. It is a major drawback of the paper. Can authors of the study continue calculations and add their results during the revision?
4- Equations governing FF, Jsc and PCE et are missing. Why?
5-How were the spectroscopic life times obtained (data in Table 2)? A detail discussion is necessary.
Author Response

(The authors gave the same response as above.)

Round 2
Reviewer 1 Report
The authors have answered the questions raised by the reviewers, and the article can be accepted